# Diagnostic and Clinical Manifestation Differences of Glucose Transporter Type 1 Deficiency Syndrome in a Family with SLC2A1 Gene Mutation

**DOI:** 10.3390/ijerph19063279

**Published:** 2022-03-10

**Authors:** Weronika Pawlik, Patrycja Okulewicz, Jakub Pawlik, Elżbieta Krzywińska-Zdeb

**Affiliations:** 1Department of Pediatric Hematooncology and Gastroenterology, Pomeranian Medical University, 71-252 Szczecin, Poland; 2Department of Pediatrics, Endocrinology, Diabetology, Metabolic Diseases and Cardiology, Pomeranian Medical University, 71-252 Szczecin, Poland; patrycja.okulewicz@gmail.com (P.O.); zdebela@op.pl (E.K.-Z.); 3Department of Orthopaedics, Traumatology and Orthpaedic Oncology, Pomeranian Medical University, 71-252 Szczecin, Poland; jakubpawlik13@gmail.com

**Keywords:** glucose transporter type 1 deficiency syndrome, pediatrics, genetic disease, rare disease, neurology

## Abstract

Glucose transporter type 1 deficiency syndrome is a rare genetic disease that manifests neurological symptoms such as mental impairment or movement disorders, mostly seen in pediatric patients. Here, we highlight the main symptoms, diagnostic difficulties, and genetic correlations of this disease based on different clinical presentations between the members of a family carrying the same mutation. In this report, we studied siblings—a 5-year-old girl and a 6-year-old boy—who were admitted to a pediatric ward with various neurological symptoms. Different diagnostic procedures such as lumbar puncture, electroencephalography, and MRI of the brain were performed on these patients. Whole genome sequencing identified mutations in the *SLC2A1* and GLUT1-DS genes, following which a ketogenic diet was implemented. This diet modification resulted in a good clinical response. Our case report reveals patients with the same genetic mutations having distinctive clinical manifestations.

## 1. Introduction

Glucose transporter type 1 deficiency syndrome (GLUT1-DS) is a rare congenital disease, which is caused by a heterozygotic or, less often, a homozygotic *SLC2A1* gene mutation, located on the p arm of chromosome 1, region 34.2 [1]. This mutation reduces the expression of the main blood–brain barrier glucose transporter—a uniport protein—and leads to energy deficits in neurons and impaired development, and function of brain [2]. Low glucose levels and lactate concentrations have been found in the cerebrospinal fluid (CSF) of these patients. A low CSF/serum glucose ratio (<0.6 in benign GLUT1-DS and <0.35 in the classical presentation) is one of the most important parameters in diagnosing GLUT1-DS [3,4]. Genetic testing for identifying mutation in the *SLC2A1* gene remains the gold standard for diagnosis [3].

Since the publishing of the first case in 1991, GLUT1-DS has been reported about 500 times worldwide, with a prevalence of 1.65–2.22 per 100,000 births [5]. De novo mutations are responsible for about 90% of GLUT-1 deficiencies, and the disease is less often transmitted in an autosomal dominant or recessive pattern [4,6]. Therefore, most of the patients have no family history of the disease. Heterozygotic parents may present a benign phenotype, or no symptoms at all [7]. Typical GLUT1 deficiency syndrome manifestations are neurological symptoms (polymorphic epileptic seizures, deceleration of head growth, impaired neurological development) and movement disorders (dystonia), which are aggravated due to caloric deficit and intense catabolism, resulting in hypoglycemia and thus impairing the psychophysical development of patients [3]. Implementing a ketogenic diet is shown to provide ketone bodies as an alternate source of energy for brain metabolism, and it remains the gold standard to date. Some reports suggest using alpha-lipoic acid and triheptanoin for treating GLUT-1 [8].

In this study, we present an unusual case of GLUT1 deficiency syndrome wherein three patients from the same family had different manifestations. The patients were the father and his two children, and we also included the mother having mental impairment of an unknown cause. In addition to showcasing the diverse clinical manifestations, we attempted to address the shortcomings in establishing the diagnosis.

## 2. Presentation of Cases

### 2.1. Case 1

A girl who is now 5 years old (born via a cesarean section at 38 weeks of gestation, second pregnancy, second delivery, birth weight of 2770 g, 10/10/10 points in Apgar score) had no symptoms during birth and in the following 12 months. At the age of 2, she experienced deterioration of speech and mobility, decreased muscle tone, microcephaly, and episodes of “absence” with no loss of consciousness multiple times throughout the day. In the neurology department, she was diagnosed with generalized seizure based on neurological examination and abnormal EEG (electroencephalography) test results and treated with valproic acid. During her second hospitalization in the neurology department, apart from the follow-up examination, her CSF was examined, and she had a low CSF/serum glucose ratio (0.41). This result prompted further genetic testing that eventually identified GLUT-1 deficiency syndrome caused by the pathogenic mutation p. Arg212His in the *SLC2A1* gene. The EEG examination and Magnetic Resonance Imaging (MRI) of the brain performed after the valproic acid treatment was administered revealed no abnormalities.

### 2.2. Case 2

A boy who is currently 6 years old (born via a cesarean section at 40 weeks of gestation, first pregnancy, first delivery, birth weight of 3360 g, 7/8/8 points in Apgar score) presented breathing difficulties, polycythemia, dysmorphic face (decreased bitemporal diameter, flat nose base, small mandible), decreased muscle tone, circulatory system disorders, and bilateral hearing impairment after birth. All of the TORCH group infections (Toxoplasmosis, Other Agents, Rubella, Cytomegalovirus, Herpes Simplex) and neuroinfection were excluded. Newborn screening was negative for inborn congenital errors of metabolism. In his third week of life, the boy was admitted to a pediatric department to diagnose decreased muscle tone. During hospitalization, a weak sucking reflex and oral feeding intolerance were also observed; thus, partial parenteral feeding was administered. The boy’s psychomotor development was impaired, and he did not reach the infant milestones (no head lifting, no rolling over, negative traction response). Physiotherapy was started in the fourth month of life with the Bobath concept and Vojta therapy. Cerebral ultrasound revealed an abnormally wide fourth ventricle of the brain, as well as hyperechogenic white matter surrounding the ventricle. MRI of the brain was performed, which showed irregularities characteristic of Dandy–Walker syndrome. At the age of 3 years, the boy had two episodes of generalized tonic-clonic seizures with hypersalivation and trismus in one month. The electroencephalography (EEG) test suggested the presence of focal lesions in the right centrotemporal area. Valproic acid was administered which provided satisfactory results.

After diagnosing the patient’s sister with GLUT1-DS, the boy’s (age of 4 years at the time) CSF was examined. A borderline low glucose concentration (42.89 mg/dL), CSF/serum glucose ratio of 0.59, and low CSF lactate level (9.10 mg/dL) were observed. The cerebrospinal fluid examination results from both patients are presented in Table 1.

The final diagnosis of glucose transporter type 1 deficiency syndrome was established based on whole exome sequencing (WES), where a heterozygous, likely pathogenic variant in the *SLC2A1* gene (p. Arg212His) was identified. A control EEG examination in the fifth year of life presented no abnormalities.

### 2.3. Treatment

After the final diagnosis of GLUT1-DS, a ketogenic diet for both patients was started based on KetoCal, along with valproic acid treatment. At first, a ketogenic ratio of 3:1 was used, but ratio adjustment was required due to an inadequate level of serum ketone bodies. The patients’ general condition and symptoms fluctuated over the course of the treatment and depended on the level of ketone bodies in their serum. Changes in serum ketone body levels and patients’ clinical presentation are shown in Figure 1 and Figure 2.

Both the boy and the girl showed good tolerance to the diet with a ratio of 2.7:1 and 2.5:1, respectively. No new symptoms were observed, and the serum glucose and acid–base balance returned back to the normal range. Since a ketogenic diet had good outcomes in both patients, valproic acid treatment was discontinued.

### 2.4. Patients’ Family History

The father’s final diagnosis of GLUT1-DS was made alongside the siblings. As a child, he presented symptoms of hypotony and generalized seizures. Symptoms disappeared as he grew older, and no treatment was implemented at any time. Currently, mild intellectual disability is observed. Due to his medical history and current state, genetic testing for *SLC2A1* gene mutation was performed, yielding a positive result. In the family, other neurological symptoms are also present. The mother suffers from mental impairment and psychological disorders, and her brother presents a decreased muscle tone. Both have never been tested for *SLC2A1* gene mutation (Figure 3). The study conducted by S. Girirajan indicated that the phenotypic variation in some neurological genomic diseases may be associated with the coexistence of additional large variants that are more likely transmitted by females [9]. The importance of these data remains inconclusive in our case, as the mother refused to be tested.

## 3. Discussion

Glucose transporter type 1 deficiency syndrome appears to be a challenging issue for many healthcare professionals—mostly due to its diverse nature of symptoms and clinical course. Therefore, now, mostly the term “*SLC2A1-*related disorders”—a phenotype including various epilepsy syndromes and motor disorders (e.g., type 9 and type 18 dystonia)—is used [10]. Typically, the first symptoms are observed in the first year of life. The most characteristic are seizures (most often the absence type), psychomotor development delay, ataxia, dysarthria, muscle tone abnormalities, and mental disability [11]. Some of the less characteristic symptoms might include confusion, lethargy, sleep disorders, and headaches. Acquired microcephaly is also a characteristic trait in patients’ phenotype [12]. GLUT1-DS is responsible for more than 10% of absence seizures in children below the age of 4 years, whereas only 10% of patients suffering from GLUT1-DS experience no seizures at all [13,14]. Pathological activity in children during caloric deficit, before meals, during stress, and after physical workout should also be of interest to a physician.

Lumbar puncture and CSF analysis are crucial in the diagnosis of GLUT1-DS. Low glucose levels (with normal serum glucose) and low lactates in CSF are typical for this syndrome [15]. Currently, the upper limit of CSF glucose is considered to be 40 mg/dL (2.2 mmol/L), but a recently published systematic review (with 147 patients included) proved the CSF glucose levels of GLUT1-DS patients to vary between 16.2 and 50.5 mg/dL. A CSF/serum glucose ratio of 0.19–0.59 and CSF lactates between 5.4 and 13.5 mg/dL were also described [16].

Analyzing the patient’s genome is currently the gold standard in diagnosing GLUT1-DS—in most cases, a pathogenic variant in the *SLC2A1* gene is identified. More than 150 mutations in the *SLC2A1* gene leading to GLUT1-DS have been diagnosed thus far [11]. The most common pathogenic mutations are Asn34, Gly91, Ser113, Arg126, Arg153, Arg264, Thr295, Arg333, Arg93, Arg212 (the mutation diagnosed in the presented case), Gly130, Ala155, and Arg330 [3,15]. Collecting a detailed family medical history is also important; for some, severe symptoms in relatives might indicate a genetic background of the disease and eventually lead to performing additional molecular testing [17]. Furthermore, diagnosing a child might result in making a similar diagnosis in the parent, who was not properly diagnosed before. However, such a familial occurrence is rarely described in the literature—there are only a handful of cases described including ours [13,18,19,20]. Yet, there are some patients suffering from GLUT1-DS (diagnosed based on the clinical presentation and CSF analysis) with no genetic abnormalities in the *SLC2A1* gene [7].

During the process of diagnosis, an EEG examination is usually performed—in most cases, the result is correct. This is because, in GLUT1-DS, the EEG result between the seizures often shows no abnormalities [21]. On the other hand, the morphology of EEG waveforms recorded in the time of a seizure may show focal slow-wave activity, high voltage spikes, or spike-and-wave patterns [22]. An important feature of EEG examination in GLUT1-DS patients is the fact that ingesting a meal induces normalization of the brain’s bioelectric activity [23]. Such a characteristic trait observed may be a useful hint for clinicians on the diagnostic path.

Radiological imaging is used to assess brain tissue and exclude diseases other than GLUT1-DS, which might also induce epilepsy in children. Studies on large populations showed patterns characteristic for GLUT1-DS in MRI and Positron Emission Tomography (PET) of the brain [24].

Though GLUT1-DS patients may present a wide spectrum of symptoms, thus far, it is believed that a specific genetic mutation (nonsense/missense mutation, or even a specific amino acid replacement) induces a specific symptom, regulates its intensity, or conditions a response to a ketogenic diet treatment [11,17]. On the other hand, there are reports which claim that there is no correlation between the spot or type of the mutation and the clinical course of the disease [4,25,26]. Our paper depicts a very unusual case of two patients sharing the same genetic mutation, but showing a different phenotype, which supports this theory. The boy mainly manifests a decreased muscle tone, movement disorders, and a dysmorphic facial feature. Moreover, he has a history of tonic-clonic seizures, and images of his brain showed various pathologies which might suggest a diagnosis different from GLUT1-DS. His sister mostly manifests absence seizures, microcephaly, and motor and speech delay. Her CSF analysis revealed characteristic abnormalities, while her brother’s result was only near the lower limit of the reference range. In both children, reducing the intake of simple sugars and implementing a ketogenic diet returned satisfactory results. The difference in symptoms presented in members of the same family, sharing the same genetic mutation, might suggest that there are other mechanisms potentially regulating the pathophysiology of this complex disease and affecting the phenotype [9].

Currently, the gold standard of treatment in glucose transporter type 1 deficiency syndrome is the implementation of a ketogenic diet, which provides ketone bodies as an alternative source of energy for the brain cells. Finding the right ketogenic proportion—that is, the proportion of fatty acids to carbohydrates and proteins—seems to be the key to achieving therapeutic success. After administering the therapy, epileptic seizures often become less frequent, and the patients’ general condition improves. Additionally, the ketogenic diet shows a neuroprotective effect [27]. Administering this diet in adults manifesting a benign clinical course might also be worth considering; however, to this day, there are no reliable studies proving the usefulness of such a treatment.

## 4. Conclusions

Familial glucose transporter type 1 deficiency syndrome should be considered while diagnosing recurrent epilepsies, especially those of a diverse nature and accompanied by psychomotor disabilities. Despite sharing the same genetic mutations, different phenotypes might be manifested. The key to diagnosing GLUT1-DS is examining the glucose and lactate levels in the CSF as well as introducing *SLC2A1* gene molecular testing. It is crucial for the patients’ development and wellbeing to implement an adequate ketogenic diet as soon as possible, as it is currently the only treatment providing therapeutic results.

## Figures and Tables

**Figure 1 ijerph-19-03279-f001:**
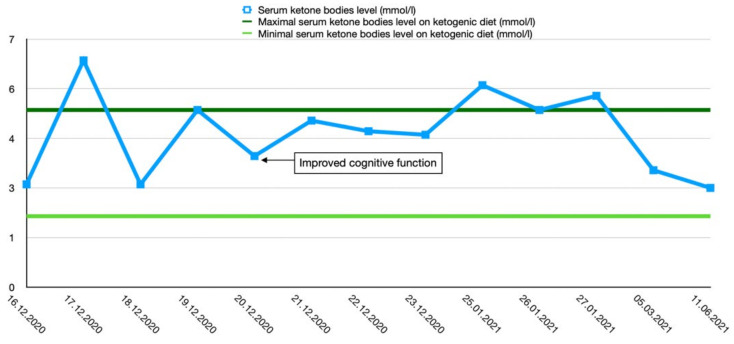
Serum ketone body level changes in female patient 1.

**Figure 2 ijerph-19-03279-f002:**
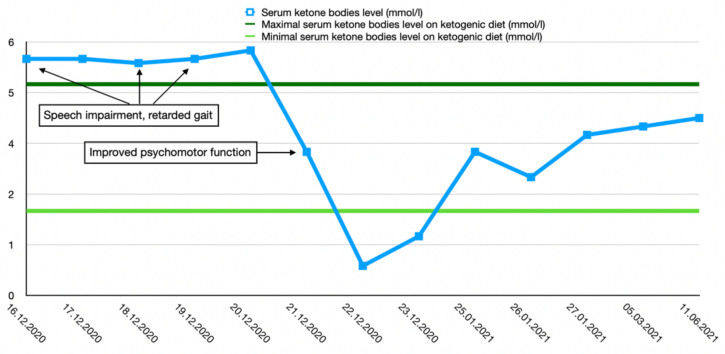
Serum ketone body level changes in male patient 2.

**Figure 3 ijerph-19-03279-f003:**
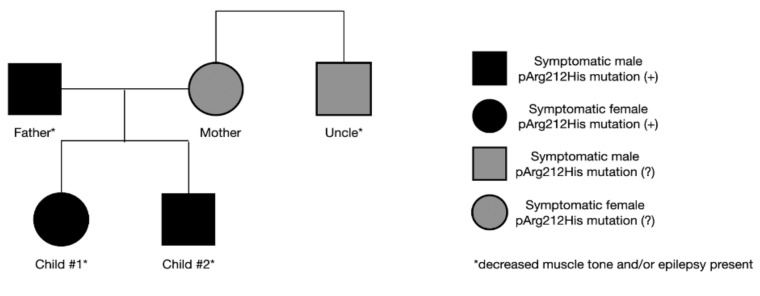
Symptoms and the presence of genetic mutation in the family.

**Table 1 ijerph-19-03279-t001:** Cerebrospinal fluid (CSF) examination in both patients.

Parameter	Reference Range	Changes Typical in GLUT1-DS	Female Patient	Male Patient
CSF glucose	48–85 mg/dL	<40 mg/dL	No Data	42.8 mg/dL
CSF lactates	10–29 mg/dL	5.4–13.5 mg/dL	No Data	9.1 mg/dL
CSF/serum glucose ratio	0.55–0.75	0.19–0.59	0.41	0.59

GLUT1-DS—Glucose transporter type 1 deficiency syndrome.

## Data Availability

Not applicable.

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
