# Peer review of "Diagnostic and Clinical Manifestation Differences of Glucose Transporter Type 1 Deficiency Syndrome in a Family with SLC2A1 Gene Mutation"

_ijerph, 2022, doi:10.3390/ijerph19063279_

Round 1

Reviewer 1 Report

Dear Authors,

The case study is well planned and informative. The cases were enrolled diligently, and they were followed up appropriately. The documentation of the cases is comprehensive, however the language or the flow of the paper isn’t up to the standards. Please incorporate the changes suggested to improve the manuscript.

Title: line 4- ‘a’ instead of one

Line 16-26: rephrase the abstract:

Here, we highlight the main symptoms, diagnostic difficulties and genetic correlations of this disease based on different clinical presentations of this disease between the members of a family carrying the same mutation. In this report we studied siblings, a 5-year-old girl and a 6-year-old boy who were admitted to a pediatric ward with various neurological symptoms. Different diagnostic procedures such as lumbar puncture, electroencephalography, and MRI of the brain were performed in these patients. Whole genome sequencing identified mutations in SLC2A1 and GLUT1-DS genes, following which a ketogenic diet was implemented. This diet modification resulted in a good clinical response. Our case report reveals patients with the same genetic mutations having distinctive clinical manifestations.

Line 31 rephrase: ‘which is caused by heterozygotic’….

Line 34-42 rephrase: This mutation reduces the expression of the main blood brain barrier glucose transporter- an uniport protein, and leads to energy deficits in neurons, impaired development, and function of brain. Low glucose levels and lactate concentration were demonstrated in the cerebrospinal fluid (CSF) of these patients. Low CSF/serum glucose ratio (<0,6 in benign GLUT1-DS and <0, 35 in classical presentation) is one of the most important parameters in diagnosing GLUT1-DS. Genetic testing for identifying mutation in the SLCA1 gene remains the gold standard for diagnosis.

Line46: delete ‘all’ GLUT-1 deficiencies and it is less often transmitted in an autosomal….

Line 52-60: replace intensify during with ‘aggravates due to caloric deficit and intense catabolism resulting in hypoglycemia and thus impair psychophysical development of patients. Implementing a ketogenic diet is shown to provide ketone bodies as an alternate source of energy for brain metabolism and it remains the gold standard to date. Some reports suggest using alphaliponic and triheptanoin for treating GLUT-1.

Line 61-66: In this study, we present a usual case of GLUT1 deficiency syndrome wherein three patients from the same family having different manifestations. The patients were father and his two children, and we also included the mother having mental impairment of unknown cause. In addition to showcasing the diverse clinical manifestations we attempted to address the shortcomings in establishing the diagnosis.

Line 67 rephrase: Presentation of cases

Line 70: replace presented with ‘had’

Line 72 rephrase: In the age of 2, she experienced deterioration….

Line 75: In the neurology department she was diagnosed with generalized seizure based on neurological examination and abnormal EEG test results and treated with valproic acid.

Line 77: please mention the time of her next hospitalization and the reason for the visit. Was it a follow up visit or hospitalization due to any symptom or emergency???

Line 77-82 rephrase: During this visit her CSF was examined and she had a low CSF/serum glucose ratio (0,41). This result prompted further performing genetic testing that eventually identified GLUT-1 deficiency syndrome caused by the pathogenic mutation in ….

Line 82-84: something is missing about the treatment. Please fix.

Line 108: valproic acid was administered that gave satisfactory results.

Table 1: please mention the sex of the patient instead of numbers.

Line 116: (WES), where a heterozygous…

Line123: ..based on KetoCal along with valproic acid treatment.

Line125: was required as the ketone bodies levels were found to be inadequate in the serum.

Figure 1 and 2: Please mention the sex of the patient in the figure legend

Line 134: Both boy and girl showed good tolerance to the diet with ration 2,7:1 and 2,5:1 respectively. No new symptoms were observed, and the serum glucose and acid-base balance went back to normal range. Since ketogenic diet had good outcomes in both the patients, valproic acid treatment was discontinued.

Line 188-189: please rephrase

Line195: does the sentence intend to convey that the EEG is normal in the patient?... Please check and fix it.

Line 201: Please elaborate the point where ‘meal induces normalization of brain’s activity’. Is it an advantage or disadvantage in diagnosis?

Reviewer 2 Report

Dear Authors,

Thank you for your effort to present this family.

I have a suggestion and minor text edits.

My suggestion is to discuss possible genetic contribution from the mother. 

You presented a family with variable expressivity of autosomal dominant SLC2A1 related disorder in three affected members. Intrafamilial variable expressivity is a characteristic of many autosomal dominant disorders. In this particular family the variability may be explainable. The mental impairment in the mother may be due to chromosomal disorder i.e. a pathogenic copy number variant or another monogenic dominant disorder. The difference in the severity between case 1 and case 2 may be due to inheritance of a different genetic factor associated with neurological disorder from the mother. Neurodevelopmental disorders due to two or more distinct genetic variants have been published in the medical literature:

Girirajan S, Rosenfeld JA, Coe BP, Parikh S, Friedman N, Goldstein A. Phenotypic heterogeneity of genomic disorders and rare copy-number variants. N Engl J Med. (2012) 367:1321–31. doi: 10.1056/NEJMoa1200395

Yang Y, Muzny DM, Xia F, Niu Z, Person R, Ding Y, Ward P, Braxton A, Wang M, Buhay C, Veeraraghavan N, Hawes A, Chiang T, Leduc M, Beuten J, Zhang J, He W, Scull J, Willis A, Landsverk M, Craigen WJ, Bekheirnia MR, Stray-Pedersen A, Liu P, Wen S, Alcaraz W, Cui H, Walkiewicz M, Reid J, Bainbridge M, Patel A, Boerwinkle E, Beaudet AL, Lupski JR, Plon SE, Gibbs RA, Eng CM. Molecular findings among patients referred for clinical whole-exome sequencing. JAMA. 2014 Nov 12;312(18):1870-9. doi: 10.1001/jama.2014.14601.PMID: 25326635 

The text edits I suggest:

16   based instead of basing

23   SLC2A1 (genes always in Italics)

53   catabolism instead of katabolism

70   the American medical literature uses “weeks of gestational age” instead of HBD (hebdomas)

146-147   please specify if the genetic test was specifically for the mutation seen in his children or sequence of SLC2A1 or other

157   the preferred term is “SLC2A1 related disorders”

215   supports instead of confirms

All the best with publishing your work.

Reviewer 3 Report

Dear Authors,

I read your Diagnostic and Clinical Manifestation Differences of Glucose Transporter Type 1 Deficiency Syndrome in One Family with SLC2A1 Gene Mutation with attention and interest. I find your research very valuable. Taking into account the present knowledge, I think your work will provide useful information about  glucose transporter type 1 deficiency syndrome treatment and diagnosis. I highly recommend this manuscript to publish after minor revision. Please, follow my detailed comments.

Deatailed comments:

Case presentation

- please, provide informations about the EEG and WES part (no information about sequencing platform, sequencing strategy and bioinformatic analyses, lack of results screen).

Author Response

This manuscript is a resubmission of an earlier submission. The following is a list of the peer review reports and author responses from that submission.